# Rapid Selective Detection and Quantification of β-Blockers Used in Doping Based on Molecularly Imprinted Nanoparticles (NanoMIPs)

**DOI:** 10.3390/polym14245420

**Published:** 2022-12-11

**Authors:** César Cáceres, Macarena del Pilar Garcia Morgado, Freddy Celis Bozo, Sergey Piletsky, Ewa Moczko

**Affiliations:** 1Facultad de Ingeniería y Ciencias, Universidad Adolfo Ibáñez, Viña del Mar 2562307, Chile; 2Laboratorio de Procesos Fotónicos y Electroquímicos, Departamento de Ciencias y Geografia, Facultad de Ciencias Naturales y Exactas, Universidad de Playa Ancha, Subida Carvallo 270, Playa Ancha, Valparaiso 2340000, Chile; 3Department of Chemistry, University of Leicester, Leicester LE1 7RH, UK

**Keywords:** doping in sports, performance enhancing drugs (PEDs), β-blockers, atenolol, labetalol, molecularly imprinting nanoparticles (nanoMIPs), enzyme-linked immunosorbent assay (ELISA), “pseudo” enzyme-linked immunosorbent assay (pseudo-ELISA), dynamic analysis light scattering (DLS), transmission electron microscope (TEM)

## Abstract

Human performance enhancing drugs (PEDs), frequently used in sport competitions, are strictly prohibited by the World Anti-Doping Agency (WADA). Biological samples collected from athletes and regular patients are continuously tested regarding the identification and/or quantification of the banned substances. Current work is focused on the application of a new analytical method, molecularly imprinted nanoparticles (nanoMIPs), to detect and determine concentrations of certain prohibited drugs, such as β-blockers, in water and human urine samples. These medications are used in the treatment of cardiovascular conditions, negative effects of adrenaline (helping to relief stress), and hypertension (slowing down the pulse and softening the arteries). They can also significantly increase muscle relaxation and improve heart efficiency. The new method of the detection and quantification of β-blockers is based on synthesis, characterization, and implementation of nanoMIPs (so-called plastic antibodies). It offers numerous advantages over the traditional methods, including high binding capacity, affinity, and selectivity for target molecules. Additionally, the whole process is less complicated, cheaper, and better controlled. The size and shape of the nanoMIPs is evaluated by dynamic light scattering (DLS) and transmission electron microscope (TEM). The affinity and selectivity of the nanoparticles are investigated by competitive pseudo enzyme-linked immunosorbent assay (pseudo-ELISA) similar to common immunoassays employing natural antibodies. To provide reliable results towards either doping detection or therapeutic monitoring using the minimal invasive method, the qualitative and quantitative analysis of these drugs is performed in water and human urine samples. It is demonstrated that the assay can detect β-blockers in water within the linear range 1 nmol·L^−1^–1 mmol·L^−1^ for atenolol with the detection limit 50.6 ng mL^−1^, and the linear range 1 mmol·L^−1^–10 mmol·L^−1^ for labetalol with the detection limit of 90.5 ng·mL^−1^. In human urine samples, the linear range is recorded in the concentration range 0.1 mmol·L^−1^–10 nmol·L^−1^ for atenolol and 1 mmol·L^−1^–10 nmol·L^−1^ for labetalol with a detection limit of 61.0 ng·mL−1 for atenolol and 99.4 ng·mL^−1^ for labetalol.

## 1. Introduction

β-blockers (adrenergic antagonists) are essential and widely used medications to treat heart diseases and diverse cardiovascular pathologies in so-called β-blocker therapy [1,2,3,4]. They reduce atrial pressure, control cardiac fibrillation, and improve the heart’s function. Additionally, β-blockers have cardioprotective effects in post-myocardial infarction and heart failure. They cause a significant decrease in heart rate and strength of the heart muscle contraction. This produces a general feeling of relaxation and calm [5,6]. The effects of tranquility and stability of the body very soon found application as performance enhancing drugs (PEDs), especially in sports that require precision and stillness such as golf, shooting, and archery. Since 1985, β-blockers were considered doping and the International Olympic Committee officially added them to the list of substances prohibited in sports [7,8]. A majority of the analytical methods for the identification and/or quantification of banned substances in biological medium still suffer complex and extensive sample preparation, insufficient accuracy, and reliability. This involves multiple analytical techniques, including immunological, biochemical, and chromatography/mass spectrometry [9,10,11,12,13]. Lack of confidence in the results has been discussed among sport organizations for years [14]. Consequently, in 1999, the World Anti-Doping Agency (WADA) was established to regulate and implement laws against doping in sports at the international level. WADA has banned numerous substances and the list is updated annually.

Within the prohibited drugs exist β-blockers such as atenolol and labetalol (see chemical structures in Appendix A), which are currently proposed as the model banned molecules. These two medications are used by athletes that require improved psychomotor coordination. They are able to relieve symptoms associated with anxiety and stress, e.g., tremors, control hypertension, lower down the pressure, relax blood vessels, and slow down the heart rate [15].

One of the most common and conventional analytical methods to measure the presence and concentration of analytes in a solution offers highly selective immunoassays. They are used in the clinical diagnosis, therapy, environmental quality monitoring, agricultural/food, forensic industries to analyze, e.g., proteins, hormones, viruses, microorganisms, DNA sequences, or drugs [16,17,18,19]. Among immunoassays, the most frequently used are enzyme-linked immunosorbent assays (ELISA). ELISA is a rapid, selective, and highly target-sensitive method. However, all advantageous, the immunoassays suffer from few serious drawbacks such as low stability of reagents (the need for refrigerated transport and challenging storage), batch-to-batch (or clone-to-clone) variability, high cost and time of manufacturing, purification, maintenance of antibodies, and the ethical issues related to animal experiments. Therefore, there exists an increasing demand to design substitutes, which can replace immunoglobulin molecules in immunoassays [20,21,22]. In the last decade, implementation of different technologies and innovations in chemistry, molecular engineering, and material science have led to the rapid development of molecular imprinting of polymeric nanoparticles (nanoMIPs), so-called plastic/synthetic antibodies, as one of the most promising strategies for the mimicking the natural receptors [23,24,25,26,27]. Herein, we propose nanoMIPs and the competitive binding between free target analyte and an enzyme-labeled conjugate analyte to immobilized nanoparticles as an alternative method to ELISA, so-called pseudo-ELISA assay. The enzyme-label conjugate reveals the amount of displacement that took place using colorimetric signal allowing detection and further quantification of the compound. The idea is to make the assays cheaper and more reliable in the outside laboratory conditions. In general, MIPs reveal physical stability (resistance against mechanical stress, high pressures, and temperatures), chemical stability (ability to work in organic solvents and extreme pH values), storage (typically more than one year at room temperature without losing of performance), and low cost (polymers are inexpensive and easy to prepare). Additionally, they can be prepared for almost any target. Molecular imprinting involves complex formation between functional monomers and the target analytes (so-called templates), followed by polymerization with crosslinkers. Furthermore, the analytes are removed, leaving cavities in polymer matrix with the specific affinity to the printed targets. The presence of the cavities allows subsequent selective recognition of related structural and functional molecules [28,29,30]. Although MIPs have been used in numerous of applications, such as chemical sensors and biosensors (environmental or clinical analysis, application in extreme environment, and replacement of chemical or biological recognition elements), separation of imprinted species from the rest of the components (chromatography and membrane separation systems), and preconcentration and selective extraction of individual or group of analytes (solid-phase extraction), they still exist certain barriers and limitations related to MIPs performance. Additionally, the process of their synthesis and the polymer purification from the template is manual, long, and relatively complicated. The presented research comprises the use of a new molecular imprinting method. It involves preparation of polymeric imprinted nanoparticles, nanoMIPs, when the analytes are immobilized on a solid-phase. In 2013, for the first time, direct replacement of antibodies with synthetic receptors, nanoMIPs, so-called “plastic antibodies”, was performed. The experiments were carried out in the competitive ELISA assays used for the antibiotic, vancomycin [19]. NanoMIPs have been demonstrated as efficient and stable substitutes for natural receptors or enzymes [31,32,33,34,35,36]. Nano MIPs reveal uniform binding sites resulting from affinity-based separation on a column containing solid-phase, minimized risk of target leaching, and templates that can be reused (important for expensive molecules) and modified [23,37]. The nanoparticles can be easily modified and immobilized similar to natural receptors [33,38,39]. The main objective of this research is to continue our previous work regarding illegal use of forbidden substances in sports [18], and to develop nanoMIPs combined with pseudo-ELISA to detect and quantify specific drugs, β-blockers, which can be used both by athletes and regular patients for therapeutic purposes. Target model molecules include atenolol and labetalol, medications used in the treatment of cardiovascular abnormal conditions.

## 2. Materials and Methods

Glass beads (GB Spheriglass^®^ A) with an average diameter of 90 μm were purchased from Potters Industries LLC. Sodium hydroxide (NaOH, ≥98.5%), (3-aminopropyl) trimethoxysilane (APTMS, 97%), toluene anhydride 99.8%, glutaraldehyde (GA 50% *m*/*m*), atenolol ≥98.0%, labetalol >98.0%, dimethyl sulfoxide (DMSO), *N*,*N*’-dimethylformamide (DMF), *N*-isopropylacrylamide (NIPAm, 97%), *N*-*N*’-methylenebisacrylamide (BIS, 99%), *N*-tert butylacrylamide (TBAm, 97%), acrylic acid (AA, 99%), ammonium persulfate (APS, >99%), *N*,*N*,*N*’,*N*’-tetramethylethylenediamine (TEMED, 99%), bovine serum albumin (BSA, ≥96%), horseradish peroxide (HRP), 3,3’,5,5’-tetramethylbenzidine (TMB), *N*-hydroxysuccinimide (NHS), *N*-(3-dimethylaminopropyl)-*N*’-ethylcarbodiimide hydrochloride (EDC, ≥98%), Tween 20, phosphate buffered saline (PBS), 2-[morpholino] ethanesulfonic acid (MES), sulfuric acid (H_2_SO_4_), and potassium bromide (KBr) were purchased from Sigma-Aldrich. Acetone, ethanol, and methanol were purchased from Merck, Chile. Milli-Q water (Millipore) was used for analysis. All chemicals and solvents were of analytical or HPLC grade and were used without further purification. Microplates, Nunclon 96 microwell plates were purchased from Thermo Scientific, Concepción, Chile and Amicon centrifugal filter unit (MWCO 30 kDa) from Merck, Chile.

### 2.1. Preparation of Solid-Phase with Immobilized Templates

NanoMIPs for atenolol (nanoMIPs-A) and nanoMIPs for labetalol (nanoMIPs-L) were both prepared based on solid-phase imprinting and the protocol adapted from Moczko et al. [18]. Initially, 60 g of 90 μm GBs (see Appendix A) was mixed with NaOH 1 mol·L^−1^ and boiled for 15 min in order to react and generate hydroxyl groups on their surface (so-called surface activation) (see Appendix A). Then, a series of washes with warm water followed by milli-Q water was carried out to eliminate all the excess of NaOH. Water from the last wash was collected to verify that the pH of the GBs was similar to the pH of water and NaOH was successfully removed. To continue the preparation of the solid-phase with immobilized templates, in the next step of the protocol, beads had to be completely dry. In order to obtain that, GBs were flushed two times with acetone and placed in the oven at 60 °C to accelerate the drying process. Modification of the surface of the GBs continued with the reaction of silanization (to obtain-NH_2_ activated GBs) by incubating the beads in a solution of 2% (*v*/*v*) of APTMS in anhydrous toluene for 24 h at ambient temperature. APTMS is a coupling agent to obtain GBs with terminal -NH_2_ groups, which allows further immobilization of template molecules (see Appendix A). After the silanization reaction, the beads were flushed with acetone and methanol, and then dried by using a vacuum filtration system. Until the next step, the functionalized GBs remained stable for approximately 6 months at ambient temperature [23,27]. Silanized surface of GBs was confirmed by Fourier transform infrared (FT-IR), Nexus, model NICOLET (see Appendix A). Collected samples were dispersed in KBr and 64 measurements were made for each sample. After silanization, the GBs were incubated in a mixture of 7% (*v*/*v*) GA solution in 0.01 mol·L^−1^ PBS for 2 h and then washed carefully with milli-Q water (eight volumes) in a vacuum filtration system.

In order to immobilize the targets (atenolol and labetalol), the activated GBs were placed in two glass vials and incubated in a solution of atenolol 5 mg·mL^−1^ for nanoMIPs-A and solution of labetalol 5 mg·mL^−1^ for nanoMIPs-L in PBS 0.01 mol·L^−1^ at pH 7.4. The reaction lasted for 24 h at ambient temperature. The last step of the protocol was to carefully wash the GBs with attached templates using only milli-Q water, dry them under vacuum, and store them at 4 °C until further use.

### 2.2. Synthesis and Purification of NanoMIPs-A and NanoMIPs-L

The synthesis of the nanoMIPs, molecularly imprinted for template molecules, was conducted through a polymerization reaction at ambient temperature in a closed degassed system. The procedure was based on mixing of monomers NIPAm, TBAm (hydrophilic monomers), and AA (acidic monomer) with BIS (crosslinker), APS (initiator of the reaction), and TEMED (catalyst). The polymerization reaction was carried out following the protocol described by Moczko et al. [18]. The synthesis was performed by dissolving 39 mg of NIPAm and 2 mg of BIS in a Duran Schott bottle with 50 mL of milli-Q water. A total of 33 mg of TBAm was dissolved in 1 mL of ethanol and added to the solution, followed by 2.2 µL of AA and milli-Q water adjusting the total volume to the 100 mL of the solution. The entire mixture was sonicated for 15 min and degassed by constant, gently bubbling with nitrogen (N_2_) for 20 min. Next, 60 g of GBs functionalized with the templates, atenolol and labetalol, were weighed, placed in a Schott bottle, and degassed for 20 min using N_2_. Furthermore, the polymerization mixture was added to the flask with GBs, stirred gently, to homogenize the contents, and again bubbled for 15 min with N_2_. Then, the polymerization reaction was initiated by adding 18 μL TEMED and 600 μL of APS (60 mg·mL^−1^), and left to polymerize for 2 h at ambient temperature. After the synthesis was finished, the entire contents of the bottle (GBs and solution) was transferred to a Hypersep cartridge in order to remove impurities, unreacted monomers, and low affinity nanoparticles, and to collect high affinity nanoMIPs. The washing process was performed in a vacuum filtration system with milli-Q water at 0 °C (60 mL × 8 times). The high-affinity nanoMIPs were eluted with milli-Q water at 60 °C (25 mL × 5 times). The solutions of high-affinity nanoMIPs-A and nanoMIPs-L in water were stored at 5 °C until the use in pseudo-ELISA assay. A schematic illustration of the synthesis of nanoMIPs is presented in Figure 1.

### 2.3. Characterization of the Size and Shape of NanoMIPs

The size and shape of the nanoMIPs for β-blockers (atenolol and labetalol) was determined in water using dynamic light scattering (DLS) analyzer, from Brookhaven Instruments Corporation Ltd. (Holtsville, NY, USA) and based on images of dry nanoMIPs obtained by a transmission electron microscope (TEM), JEOL/JEM 1200 EX II (Tokyo, Japan). The size analyzed by DLS includes data of mean hydrodynamic diameter (D), polydispersity index (PDI), and standard deviation (Std dev) of the nanoMIPs dispersion in milli-Q water. Prior to measurements, nanoMIPs were preconcentrated by evaporation and constant bubbling with N_2_ to avoid their drying. DLS was performed in 1 mL of the solution of nanoMIPs in water at 25 °C. The measurements were conducted using 3 cm^3^ disposable polystyrene cuvettes.

Before TEM analysis, samples were sonicated for 5 min, and then 20 µL of the nanoMIPs dispersion was placed on a carbon-coated copper grid and dried at ambient temperature under a fume hood. After that, thin film of a sample was ready for analysis.

### 2.4. Enzyme Linked Immunosorbent Assay (Pseudo-ELISA)

The initial step to perform pseudo-ELISA included immobilization of nanoMIPs in 96-well polystyrene microplates by direct deposition of 40 μL of their preconcentrated solution into each well. The concentration of nanoMIPs was determined using a spectrophotomer UV/Vis at 198 nm and the concentration was adjusted to 0.056 mg·L^−1^ [18]. The solvent (milli-Q water) was evaporated overnight at ambient temperature in order to obtain microplates coated with nanoMIPs via simple adsorption process. These plates were used in all required experiments while control measurements were performed on the microplate wells without nanoMIPs. A further step was based on preparation of the HRP-analyte conjugates and optimization of their concentration following the methodology and protocol published by Chianella et al. [19]. HRP (10 mg) was dissolved in 0.1 mol·L^−1^ MES buffer, pH 6 (1 mL), and then activated by adding 0.4 mg of EDC, followed by 0.6 mg of NHS to the solution. The reaction was carried out for 15 min at ambient temperature. Then, the buffer was removed by ultrafiltration using an Amicon Millipore Ultra centrifugal filter unit (30 kDa MWCO). Activated HRP was collected from the ultrafiltration unit and directly incubated with target analytes (10 mL, 1 mg·L^−1^ of atenolol and labetalol) in PBS buffer at pH 7.4 for 2 h. In order to remove unreacted molecules, the HRP-atenolol conjugate (HRP-A), and the HRP-labetalol conjugate (HRP-L), 10 washes with PBS (5 mL) were performed using an Amicon Millipore Ultra centrifugal filter unit (30 kDa MWCO). Finally, the conjugate was dissolved in 2 mL of milli-Q water and its concentration was estimated by comparison with the enzymatic activity of the free enzyme. It was then stored frozen at −18 °C and used as the stock solution in the pseudo-ELISA measurements.

Optimization of the concentration of both HRP-analyte conjugates (HRP-A and HRP-L) was performed under the previously optimized conditions using different concentrations of HRP-conjugates [19]. Before actual measurements, microplates were conditioned by washing each well with 0.01 mol·L^−1^ PBS (2 × 250 µL), followed by 1 h incubation with the blocking solution (300 µL of 0.01 mol·L^−1^ PBS containing 0.1% of BSA and 1% of Tween 20). After further treatment with 0.01 mol·L^−1^ PBS (3 × 250 µL), 100 µL of HRP-A, and HRP-L, conjugates were added using several different dilutions (1:200, 1.400, 1:600, 1:800, 1:1000, 1:1200, and 1:1400). Both microplates were kept in the dark for 1 h at ambient temperature. After they were washed with 0.01 mol·L^−1^ PBS (3 × 250 µL), HRP substrate and TMB reagent (100 μL) was added to each of the microplate wells and incubated for exactly 10 min. After that, by adding 100 μL H_2_SO_4_ (0.5 mol·L^−1^), the enzymatic reaction was stopped. The control measurements were performed by coating the microplates wells only with analytes, without nanoMIPs-A and nanoMIPs-L. The suitable concentration of the conjugate was determined by the greatest difference in color development between controls and wells with nanoparticles. The absorbance of the content of each well was measured at 450 nm using a UV/Vis microplate reader.

Following the optimization of the assay conditions, the next step of the analysis was to perform competitive-type ELISA replacing natural antibodies with produced nanoMIPs to determine and quantify atenolol and labetalol in the samples. Binding competition was made between the free analyte and the HRP-analyte conjugate using immobilized receptors (nanoMIPs). The analytical signal depended on the colorimetric reaction between the enzyme (HRP) and the enzyme substrate (chromogenic reagent, TMB). The principle of competitive pseudo-ELISA immunoassay is presented in Appendix A.

The protocol was adopted from Moczko et al. [18]. Microplate wells were coated with nanoMIPs by dispensing undiluted stock solution (40 μL, 0.056 mg·mL^−1^) into each well followed by overnight evaporation. Each well was conditioned by washing with 0.01 mol/L PBS (2 × 250 μL) followed by 1 h incubation with blocking solution, 0.01 mol·L^−1^ PBS (300 μL) containing BSA (0.1%) and Tween 20 (1%). Wells were then washed with 0.01 mol·L^−1^ PBS (3 × 250 μL). To each well, 100 μL of selected dilution of HRP-A or HRP-L was added, containing different concentrations of a free analyte standard solution, atenolol and labetalol. Plates were incubated in the dark for 1 h at ambient temperature. Then, they were washed with 0.01 mol·L^−1^ PBS (3 × 300 μL) containing BSA (0.1%) and Tween 20 (1%), followed by addition of TMB reagent (100 μL). After 10 min incubation, the enzymatic reaction was stopped by the addition of 100 µL H_2_SO_4_ (0.5 mol·L^−1^). Color change was determined by measuring the absorbance of each well at 450 nm using a UV/Vis microplate reader.

### 2.5. Analysis of Atenolol and Labetalol in Urine Samples

The next step of the work was to demonstrate the assay capability to detect atenolol and labetalol in real, biological media, particularly in human urine samples, the most common matrix to analyze forbidden substances in sports. Human urine was collected from a healthy person over 18 years old and all protocols used in the experiments were approved by an institutional committee of Bioethics from the University of Concepcion (Chile). In further experiments, the stock solution of urine was diluted (1:1, 1:10, 1:100, and 1:1000), decanted, and spiked with known concentrations of analytes.

### 2.6. Competitive Pseudo-ELISA Assay to Analyze cross Reactivity of the NanoMIPs-A and NanoMIPs-L

The final step of the analysis of the assays was performed in order to confirm pseudo-ELISA affinity and selectivity. NanoMIPs-A and nanoMIPs-L were tested in human urine samples and their binding was evaluated based on the interaction with molecules that were similar in structure or molecular weight to atenolol and labetalol. The ultimate test was performed using different type of the nanoparticles (control nanoMIPs prepared for another template), which were immobilized on the microplate and tested with target β-blockers.

## 3. Results and Discussion

### 3.1. Synthesis and Characterisation of NanoMIPs-A and NanoMIPs-L

After atenolol and labetalol were immobilized on the solid-phase, the polymerization of nanoMIPs was performed in milli-Q water following known and previously tested protocol [18,40]. The polymerization of nanoMIPs was based on the electrostatic, hydrophobic, and hydrogen bonding interactions between target molecules and the mixture of various monomers used in the reaction. The synthesis of nanoMIPs was carried out for 2 h in the presence of different polymerization monomers (NIPAm, BIS, TBAm, and AA) and the GBs with immobilized target molecules. The radical polymerization reaction began by adding the initiation mixture (APS and TEMED). After 2 h, nanoMIPs were washed gently at 0 °C to remove fragments of polymers, monomers, impurities, and low affinity nanoparticles without drying the beads with the templates. The high affinity nanoparticles were eluted at 60 °C. A schematic illustration of the synthesis of nanoMIPs is presented in Figure 1. The high affinity nanoparticles were collected and stored until further use.

The size of nanoMIPs was examined by a dispersion analyzer, DLS. Samples were sonicated for 5 min and the measurements were carried out in water at 25 °C. The results of the nanoparticles’ hydrodynamic diameters (D) and the polydispersity indexes (PDI) are presented in Figure 2. The mean size for nanoMIPs-A was 248 nm with an average polydispersity of the sample PDI 0.31 and for nanoMIPs-L 267 nm with PDI 0.27. The low polydispersity indicates homogeneous size of nanoMIPs distribution.

The size and shape of synthesized nanoMIPs was also analyzed by microscopy technique, TEM. The results demonstrated spherical shape of nanoparticles around 300–400 nm for both nanoMIPs-A and nanoMIPs-L. Images are shown in Figure 3. There can be seen slightly different sizes of nanoparticles by using two different techniques. A bigger size in the case of a TEM analysis can be explained by hydrophobicity of the nanoparticles and perhaps presence of a film of other hydrophobic impurities or monomers present in the aqueous solution. The results were very interesting and worth further study. Therefore, after 3 months of storing the nanoMIPs at 5 °C, the original samples were analyzed again using TEM. It was observed that the nanoMIPs tended to aggregate and lose their initial spherical shape and size (see Figure 4). This suggested that in time, nanoMIPs might get decomposed or dissolved.

### 3.2. Testing the Affinity and Selectivity of NanoMIPs-A and NanoMIPs-L Using Pseudo-ELISA Assay

After determination of the physical characteristics of nanoMIPs, their chemical properties related to molecular binding were investigated using competitive pseudo-ELISA assay. Experiments were performed in order to confirm nanoparticles’ ability to detect and quantify β-blockers in a solution, particularly in water and human urine. First, nanoparticles were attached into polystyrene microplate wells (40 μL of particles in milli-Q water, with previously adjusted concentration of 0.056 mg·mL^−^^1^). The immobilization of the nanoMIPs was achieved similarly to the coupling of antibodies to microplates through the process of physical adsorption. The method was sufficient, allowing nanoparticles to remain attached to the well surfaces even after several washes with PBS. Control measurements were performed in the wells of the microplates without nanoMIPs. After the procedure of immobilization, all plates were kept in dry conditions until further use. The next step was to conjugate analytes with HRP, a commonly used enzyme for colorimetric ELISA immunoassays. Therefore, HRP was coupled to the analytes, atenolol and labetalol, to form the HRP-A and HRP-L conjugates. Furthermore, the suitable concentration of HRP-analyte conjugates was optimized in order to find the best conditions for the final competitive assay. The stock solution was diluted from 1:200 to 1:1400 fold. After preparing microplates, containing immobilized nanoMIPs and controls, each dilution of the conjugate was added into the wells and incubated for 1 h at ambient temperature in the dark. The results were analyzed at 450 nm and the selection of the dilution was based on a color development during colorimetric reaction with the chromogenic agent TMB, a well-known substrate for HRP. The results indicated a suppression of non-specific binding of the HRP-A and HRP-L conjugates to the non-coated wells, and the specific bonds between the HRP-conjugates and the immobilized nanoparticles, nanoMIPs. The determination of an optimal dilution of HRP-analyte conjugate corresponded to the ratio between microplate wells with nanoMIPs and the controls (see Appendix A). The largest difference was observed for 1:200 dilution and therefore it was selected as the optimal to use in subsequent experiments. The ratio for atenolol was 10.10 and labetalol 3.14 (see Appendix A). Further experiments were performed to demonstrate the affinity and selectivity of synthesized nanoMIPs for a given analytes. In order to prove their viability, competitive pseudo-ELISA assay was performed both in water and human urine samples. Previously prepared microplates with coated nanoMIPs-A, nanoMIPs-L, and the controls (blanks without coated nanoparticles) were conditioned by using previously developed protocol by Chianella et al. [19]. Microplates wells were washed with PBS (2 × 250 μL) followed by an incubation for 1 h with blocking solution containing PBS (300 μL), BSA (0.1%) and Tween 20 (1%). Wells were then washed with PBS (3 × 250 μL). To each well, a solution of HRP-A and HRP-L was added (100 μL from selected stocks), containing free atenolol and labetalol. Plates were incubated in the dark at ambient temperature for 1 h, when the free analyte and the analyte conjugated with HRP were competing for the nanoMIPs active site. After that, the solution and unreacted molecules were removed and the wells were washed with PBS (300 μL), BSA (0.1%), and Tween 20 (1%), followed by addition of the TMB reagent (100 μL). Control experiments were performed using uncoated wells. The results were analyzed based on the color development after the chemical reaction with TMB. The results were analyzed by measuring the absorbance of each well at 450 nm. After 10 min incubation, the enzymatic reaction was stopped by the addition of H_2_SO_4_ (0.5 M, 100 μL). The affinity and linearity of the assay is presented in Figure 5A,B for atenolol and labetalol, respectively.

The ELISA assay revealed linear response in the concentration range 1 nmol·L^−1^–1 mmol·L^−1^ for atenolol and 1 mmol·L^−1^–10 mmol·L^−1^ for labetalol. It can be seen that nanoMIPs-A (Figure 5A) demonstrated higher absorbance, larger linear range, and operational ability in the lower concentrations of free analyte. Results obtained for nanoMIPs-L were slightly different (Figure 5B). The linear range of the assay was significantly reduced and covered only higher concentrations of free labetalol. Therefore, they suggested lower affinity of the nanoMIPs-L to labetalol. Following experiments in water, the next set of measurements involved developing and testing pseudo-ELISA assay in human urine samples. Prior to any analysis, in order to minimize interferences from the urine, reduce their potential negative contribution, and maximize the sensitivity of the method, all samples were filtrated. First, the concentrated stock solution of urine was diluted (1:1, 1:10, 1:100, and 1:1000) and then decanted and filtrated by using 0.22 μm syringe filter (PVDF). The concentrations of analytes in urine samples were determined following the results of the competitive assay used in water, based on absorbance and the calibration curve of a competition between HRP-A/free atenolol and HPR-L/free labetalol. Additionally, control experiments were performed using uncoated microplates (blanks). In all cases, the nanoMIPs assay prepared for atenolol and labetalol revealed higher binding compared with blanks. Based on the intensity of absorbance, the best solution was chosen for 1:100 and 1:1000 dilutions (see Figure 6).

The calibration curve for the detection of atenolol and labetalol in urine was obtained using two selected dilutions. Results are demonstrated in Figure 7. The same as for experiments in water, graphs show calibration curves prepared based on the binding of nanoMIPs at different concentrations of free analytes (nanoMIPs-A/atenolol and nanoMIPs-L/labetalol). In all experiments, measurements were obtained using the same dilution of HRP-analyte conjugate; therefore, the free analyte and the analyte marked with the HRP were competing for a binding side of nanoMIPs. The results indicate that the absorbance was associated with the concentration of free analyte. The competitive ELISA assay revealed linear response in the concentration range 10 nmol·L^−1^–0.1 mmol·L^−1^ for atenolol and 10 nmol·L^−1^–1 mmol·L^−1^ for labetalol. Therefore, experiments again demonstrated better affinity of the assay prepared for atenolol and much lower affinity for labetalol. In the case of nanoMIP-A, it can be observed higher absorbance and larger linear range at the lower concentrations of free analyte. Results obtained for nanoMIPs-L demonstrated weaker affinity of nanoparticles prepared for labetalol. Additionally, the linear range of the assay was observed only for higher concentrations of free labetalol.

The next step of the evaluation of the assay was to test the selectivity (cross-reactivity) of prepared nanoMIPs for β-blockers in human urine samples (see Figure 8). The experiments were performed using analytes related to atenolol and labetalol. The first molecule was octopamine, which is also a banned molecule in sports. The other selected molecule was pseudoephedrine, a molecule with similar molecular weight. Figure 8A,C shows the results of the selectivity of the nanoMIPs-A and nanoMIPs-L for the different analytes compared with their actual targets (atenolol and labetalol). It clearly indicates negligible binding of the analogue analytes to the nanoMIPs prepared for atenolol or labetalol. Additional experiments were performed using nanoparticles prepared for different target (serotonin) and tested with the same series of analytes. The results are reported in Figure 8B,D. It can be seen a non-significant interaction between serotonin nanoMIPs and different template molecules. Analysis of variance ANOVA for the presented data is included in Appendix A.

## 4. Conclusions

This paper is a continuation of our research on the detection and quantification of the prohibited substances in sports, anti-DOPING [18]. It describes in detail the synthesis and characterization of nanoMIPs for two β-blockers atenolol and labetalol. Both substances are banned by the World Anti-Doping Agency (WADA). The nanoMIPs similar to natural antibodies were used for the detection of atenolol and labetalol in competitive ELISA assay in water and human urine samples. The nanoMIPs were synthesized using methodology of solid-phase synthesis with immobilized template. Their size and shape were characterized using DLS and TEM. The average size of atenolol nanoMIPs was 248 nm and for labetalol 267 nm. The results obtained by TEM demonstrated the spherical shape of the nanoMIPs. Further experiments confirmed that nanoMIPs could be used in pseudo-ELISA with high specificity and sensibility for the detection of atenolol. The measurements performed using labetalol revealed high selectivity of nanoMIPs for target molecules, but lower affinity compared to nanoMIPs prepared for atenolol. The detection limits in water were 50.6 ng mL^−1^ for atenolol and 90.5 ng·mL^−1^ for labetalol. In human urine, the detection limits were 61.0 ng·mL^−1^ for atenolol and 99.4 ng·mL^−1^ for labetalol. Nevertheless, the assay demonstrated high selectivity for both molecules with very low cross-reactivity. Therefore, the proposed approach might open new possibilities to track prohibited substances in complex biological matrices. The presented approach could significantly improve and modernize the anti-doping system, providing a cost-effective and simple alternative to common analytical methods, which are nowadays used for doping control. Moreover, non-complicated synthesis of nanoMIPs and rapid analysis of pseudo-ELISA suggest that the assay can be prepared and used for several analytes in a relatively short time. Additionally, an interesting fact was that after 3 months of storage, nanoMIPs started to decompose from their original form, losing round shape and significantly decreasing their size. This might imply valuable characteristics of biodegradability of the presented nanomaterials.

## Figures and Tables

**Figure 1 polymers-14-05420-f001:**
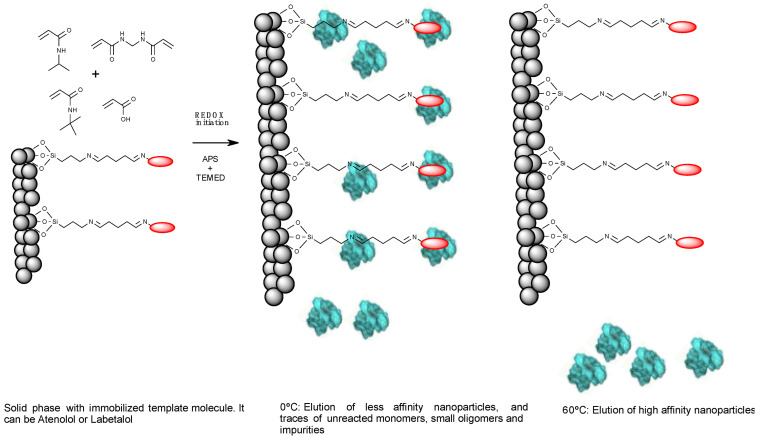
Schematic illustration of the synthesis of nanoMIPs in milli-Q water. The figure includes functionalized GBs, the mixture of monomers, the polymerization process, washing of solid-phase, and collection of high-affinity nanoparticles.

**Figure 2 polymers-14-05420-f002:**
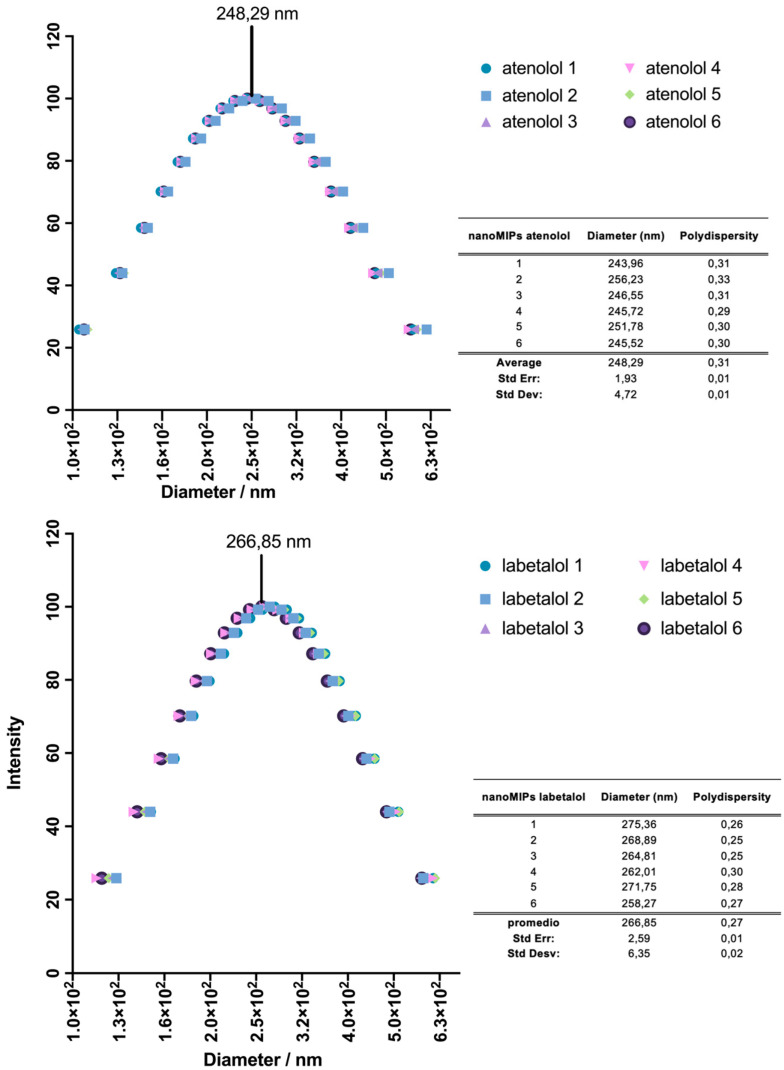
DLS graphs of the nanoMIPs-A (**top**) and nanoMIPs-L (**bottom**). On the right are tables including values of D and PDI of the nanoparticles.

**Figure 3 polymers-14-05420-f003:**
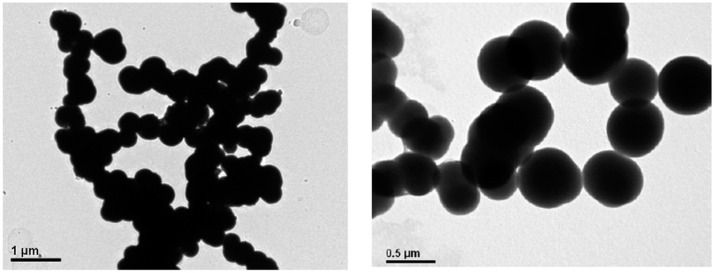
TEM images of nanoMIPs-A (**top**) and nanoMIPs-L (**bottom**). On the left are the images in the 1 μm scale and on the right 0.5 μm.

**Figure 4 polymers-14-05420-f004:**
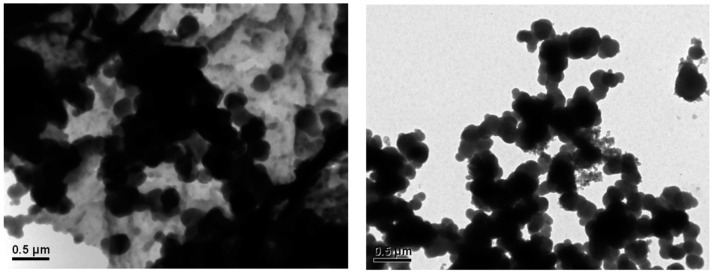
TEM images after 3 months of storing at 5 °C for nanoMIPs-A (**left**) and nanoMIPs-L (**right**).

**Figure 5 polymers-14-05420-f005:**
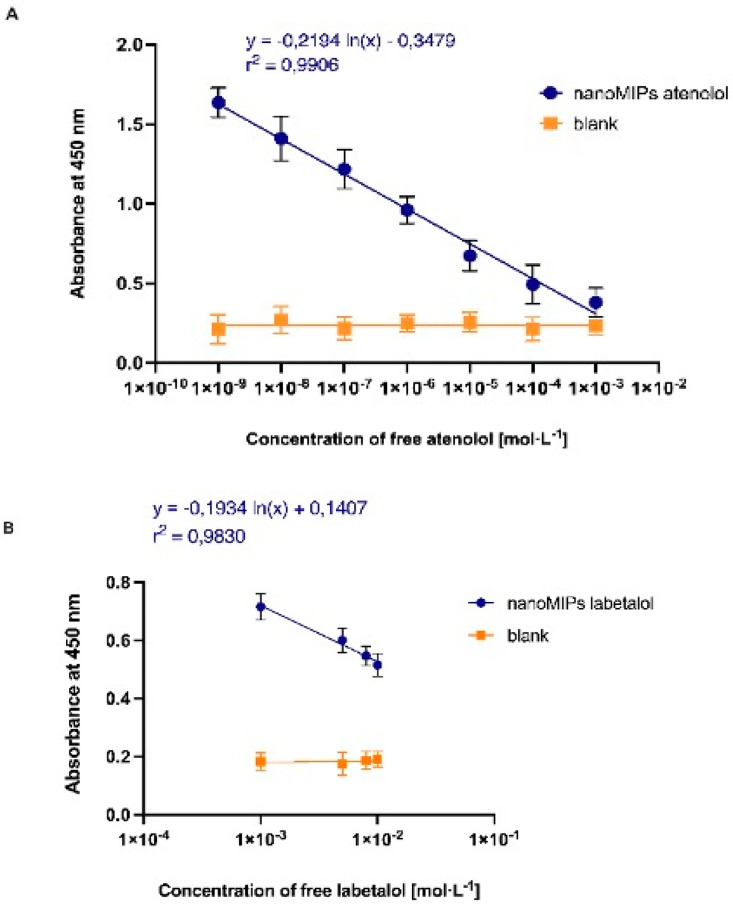
Results of competitive-type pseudo-ELISA assay for nanoMIPs-A (**A**) and nanoMIPs-L (**B**) obtained in water.

**Figure 6 polymers-14-05420-f006:**
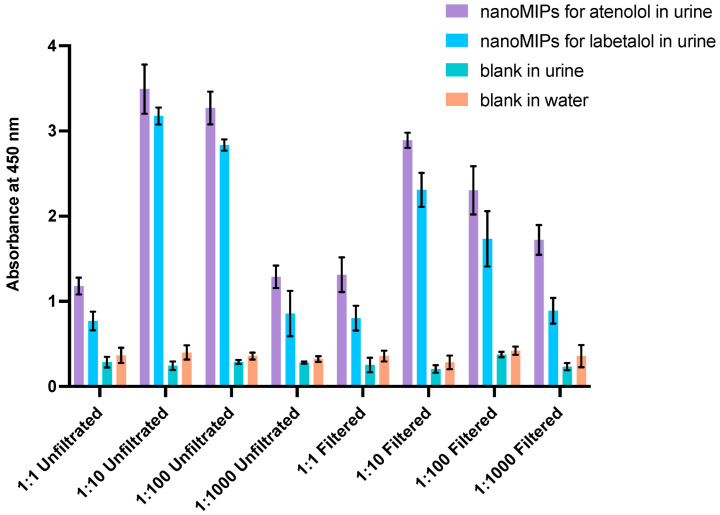
The results obtained in human urine for different dilutions of the HRP-A and HRP-L conjugates and controls. Purple bars represent absorbance corresponding to the binding of the conjugate and nanoMIPs-A, blue to the binding of the conjugate and nanoMIPs-L, and green and orange are controls without nanoMIPs.

**Figure 7 polymers-14-05420-f007:**
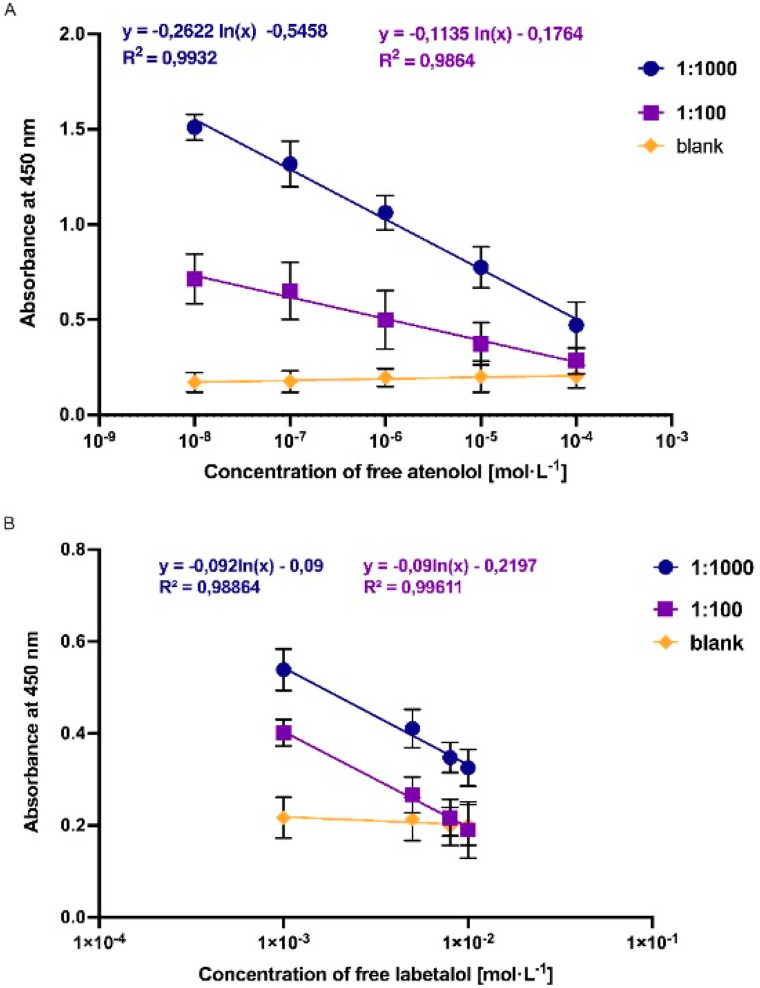
Results of competitive-type pseudo-ELISA assay for nanoMIPs-A and nanoMIPs-L obtained in water (**A**,**B**).

**Figure 8 polymers-14-05420-f008:**
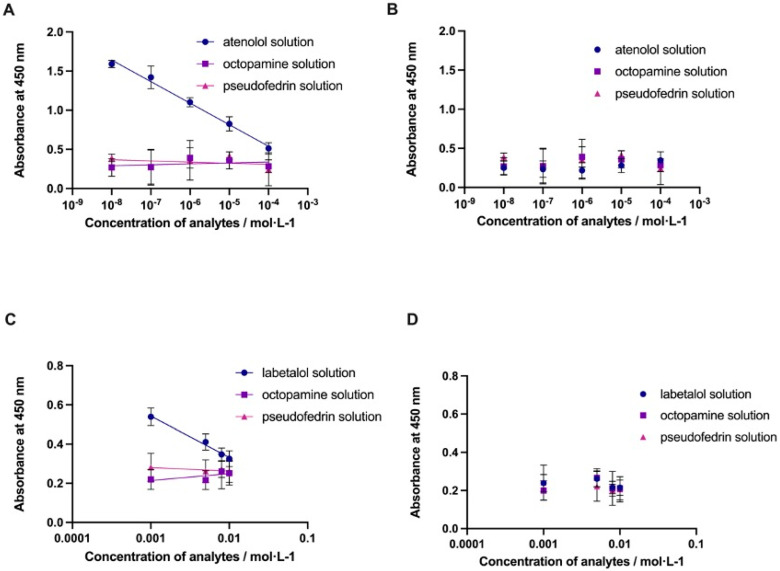
Results of competitive-type pseudo-ELISA assay for nanoMIPs-A (**A**), nanoMIPs-L (**C**), and control performed for nanoMIPs prepared for serotonine (**B**,**D**).

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
