# Peer review of "Rapid Selective Detection and Quantification of β-Blockers Used in Doping Based on Molecularly Imprinted Nanoparticles (NanoMIPs)"

_polymers, 2022, doi:10.3390/polym14245420_

Round 1
Reviewer 1 Report
Journal: Polymers
Manuscript Number: polymers-1960056
Title: Rapid selective detection and quantification of β-blockers used in doping based on Molecularly Imprinted Nanoparticles (nanoMIPs).
Recommendation: Major revision
Comments
As a reviewer, I have been checked the similarity index while review process, unfortunately, the submitted manuscript entitled “Rapid selective detection and quantification of β-blockers used in doping based on Molecularly Imprinted Nanoparticles (nanoMIPs)” by César Cáceres et al., shows similarity index of 30% (https://www.turnitin.com). Therefore, I do not recommend it present form for publication in Polymers.
Author Response
Response to Reviewers
We really appreciate your time and the effort to revised enclosed manuscript. All suggested corrections were incorporated into the text. Please find enclosed the revised version of the manuscript and below the detailed response regarding your comment.
Comment: As a reviewer, I have been checked the similarity index while review process, unfortunately, the submitted manuscript entitled “Rapid selective detection and quantification of β-blockers used in doping based on Molecularly Imprinted Nanoparticles (nanoMIPs)” by César Cáceres et al., shows similarity index of 30% (https://www.turnitin.com). Therefore, I do not recommend it present form for publication in Polymers.
Answer: We fully agree that recently submitted manuscript indeed has similarities but only to our own work (E. Moczko, R. Diaz, B. Rivas, C. Garcia, E. Pereira, S. Piletsky, C. Caceres, Molecularly Imprinted Nanoparticles Assay (MINA) in Pseudo ELISA: An Alternative to Detect and Quantify Octopamine in Water and Human Urine Samples, Polymers, 11 (2019) 1-13 or I. Chianella, A. Guerreiro, E. Moczko, J.S. Caygill, E.V. Piletska, I. Sansalvador, M.J. Whitcombe, S.A. Piletsky, Direct Replacement of Antibodies with Molecularly Imprinted Polymer Nanoparticles in ELISA-Development of a Novel Assay for Vancomycin, Anal Chem, 85 (2013) 8462-8468).The fact that the manuscript describes a continuation of our previous work it is mentioned few times in the manuscript. Following the same protocols, procedures and techniques to characterize nanoMIPs made it very difficult to avoid similar phrases regarding technical parts and experimental work. Nevertheless, corrections have been included to the new version of the manuscript in order to avoid repetitions.
Reviewer 2 Report
Comments:
The manuscript “Rapid selective detection and quantification of β-blockers used in doping based on Molecularly Imprinted Nanoparticles (nanoMIPs)” by Ewa Moczko and colleagues introduced molecularly imprinted nanoparticles (nanoMIPs) to detect and determine the concentration of certain prohibited drugs, β-blockers in water and human urine samples. This technique showed high binding capacity, affinity, and selectivity. Overall, the manuscript is well written. However, the reviewer believes that additional points of clarifications could potentially be addressed to further strengthen the manuscript.
1. In line 307, change ‘examine’ to ‘examined’
2. In line 308, the author measured the size of the nanoMIPs in water by DLS. I think the pH value of the solvent will affect the size of the nanoMIPs via DLS, the reviewer recommends the author use the 1x PH=7.4 PBS to measure the size of the nanoMIPS.
3. What about the zeta potential of the nanoMIPs? Zeta potential is also an important characterization parameter of nanoparticles.
4. In line 436, the author states that ‘It can be seen non-significant interaction between serotonin nanoMIPs and different template molecules.’ The reviewer recommends the author added the statistically analysis in Figure 8 and then says there was non-significant difference between serotonin nanoMIPs and different template molecules
Author Response
Response to Reviewers
We really appreciate your time and the effort to revised enclosed manuscript. All suggested corrections were incorporated into the text. Please find enclosed the revised version of the manuscript and below the detailed response regarding your comments.
Comment 1: In line 307, change ‘examine’ to ‘examined’
Answer 1: Thank you, the spelling has been corrected. We
Comment 2: In line 308, the author measured the size of the nanoMIPs in water by DLS. I think the pH value of the solvent will affect the size of the nanoMIPs via DLS, the reviewer recommends the author use the 1x PH=7.4 PBS to measure the size of the nanoMIPS.
Answer 2: Indeed, using water as a solvent in DLS measurements is not the ideal solution as the size might be 2–10 nm larger than actual size. However differences, our intention was to know only average size of the nanoparticles. Additionally, milliQ water was the solvent in the final elution of nanoMIPs and our intention was to analyze their size in the original state.
Comment 3: What about the zeta potential of the nanoMIPs? Zeta potential is also an important characterization parameter of nanoparticles.
Answer 3: Unfortunately, available to our use DLS does not have the module to analyze Zeta potential.
Comment 4: In line 436, the author states that ‘It can be seen non-significant interaction between serotonin nanoMIPs and different template molecules.’ The reviewer recommends the author added the statistically analysis in Figure 8 and then says there was non-significant difference between serotonin nanoMIPs and different template molecules.
Answer 4: We are very grateful for the comment. Analysis of variance ANOVA for presented data has been included in Supplementary Information S8.
Round 2
Reviewer 2 Report
The authors have addressed all my comments. I would suggest to accept the current version.
Author Response
Thank you for your comment